# A Systematic Review and Meta-Analysis on the Efficacy of Locally Delivered Adjunctive Curcumin (*Curcuma longa* L.) in the Treatment of Periodontitis

**DOI:** 10.3390/biomedicines11020481

**Published:** 2023-02-07

**Authors:** Louisa M. Wendorff-Tobolla, Michael Wolgin, Gernot Wagner, Irma Klerings, Anna Dvornyk, Andrej M. Kielbassa

**Affiliations:** 1Center for Operative Dentistry, Periodontology, and Endodontology, Department of Dentistry, Faculty of Medicine and Dentistry, Danube Private University (DPU), 3500 Krems, Austria; 2Department of Evidence-based Medicine and Evaluation, Danube University Krems, 3500 Krems, Austria; 3Department of Propaedeutics of Therapeutic Dentistry, Faculty of Dentistry, Poltava State Medical University (PSMU), 36011 Poltava, Ukraine

**Keywords:** chlorhexidine, clinical attachment level, curcumin/turmeric, mechanical debridement, periodontitis, periodontal treatment, probing pocket depth

## Abstract

This meta-analysis intended to assess evidence on the efficacy of locally delivered curcumin/turmeric as an adjunctive to scaling and root planing (SRP), on clinical attachment level (CAL) and probing pocket depth (PPD), compared to SRP alone or in combination with chlorhexidine (CHX). RCTs were identified from PubMed, Cochrane Library, BASE, LIVIVO, Dentistry Oral Sciences Source, MEDLINE Complete, Scopus, ClinicalTrials.gov, and eLibrary, until August 2022. The risk of bias (RoB) was assessed with the Cochrane Risk of Bias tool 2.0. A random-effects meta-analysis was performed by pooling mean differences with 95% confidence intervals. Out of 827 references yielded by the search, 23 trials meeting the eligibility criteria were included. The meta-analysis revealed that SRP and curcumin/turmeric application were statistically significantly different compared to SRP alone for CAL (−0.33 mm; *p* = 0.03; 95% CI −0.54 to −0.11; I^2^ = 62.3%), and for PPD (−0.47 mm; *p* = 0.024; 95% CI −0.88 to −0.06; I^2^ = 95.5%); however, this difference was considered clinically meaningless. No significant differences were obtained between patients treated with SRP and CHX, compared to SRP and curcumin/turmeric. The RoB assessment revealed numerous inaccuracies, thus raising concerns about previous overestimates of potential treatment effects.

## 1. Introduction

Severe periodontal disease affected about 1.1 billion people globally in 2019 [1], with an overall prevalence of 67.4% (probing pocket depths between 4 and 5 mm), with 9.1% accounting for adolescents, 27.7% for adults, and 30.6% for elderly people [2]. Consequently, periodontitis can be considered a cross-generational, global public health problem with widely ranging effects such as bleeding gums, periodontal pockets, bone loss, and functional as well as aesthetic issues [3]. Furthermore, it is considered a risk factor for several systematic diseases, including cardiovascular disorders, rheumatoid arthritis, and chronic obstructive pulmonary diseases, as well as non-alcoholic liver diseases [3,4].

The primary goal of periodontal treatment is the removal of highly organized microorganisms embedded in an extracellular, polysaccharide matrix attached to the tooth’s surface [5]. Biofilm management is usually achieved by sustainable biofilm disintegration consisting of instructions on effective oral hygiene, mechanical debridement of tooth surfaces, and removal of co-factors favoring re-accumulation, together with a regular recall system in which the current periodontal situation is evaluated [6]. Local therapeutical substances such as rinsing solutions, gels, and chips are administered to increase the success of the treatment and enable long-term clinical management of periodontitis [7].

One such therapeutical substance is chlorhexidine, which is bacteriostatic and bactericidal to gram-positive and gram-negative bacteria; additionally, chlorhexidine inhibits plaque formation by having a high affinity for binding spots of bacteria [8]. Application of chlorhexidine as a standard chemotherapeutic agent can, however, be associated with several undesirable side effects. Being exposed over a longer period of time, chlorhexidine can cause brown staining of the teeth, decreased taste sensation, oral mucosal lesions, and/or increased calculus formation [8].

Another common therapeutic substance is hydrogen peroxide, which operates as a disinfecting agent by releasing oxygen, thus creating an environment that is able to inhibit anerobic bacteria growth [9]. Notwithstanding, several adverse reactions to highly concentrated hydrogen peroxide may be observed; short durations can cause erythema or mucosal sloughing, whereas application for longer periods can lead to inflammation and/or hyperplasia [10].

Investigations have been initiated to examine the use of cytostatics, photodynamic therapy, metal ions, and natural compounds/oils in inflammatory-mediated conditions [11,12,13,14,15,16]. One of these alternatives is curcumin, which is a natural constituent found in the turmeric plant [17]. In contrast to conventional therapeutic substances, curcumin intervenes in the pathophysiological process of inflammation rather than working solely as an anti-bacterial [17]. It is extracted from the turmeric plant *Curcuma longa* L. (http://www.theplantlist.org; accessed on 29 July 2022) [17], and is thought to act anti-inflammatory by inhibiting the mRNA and protein expression of Cyclo-oxygenase-2 (COX-2) [18,19], through down-regulation of NF-kB activation [20]. Further proposed anti-inflammatory methods of action are that curcumin declines the activity of phospholipase A2, C and D [21,22] and inhibits lipoxygenase [23,24,25]. Additionally, an anti-bacterial mode of action is observed by curcumin; by inserting itself into the hydrophobic cellular membrane, thus disrupting membrane integrity, curcumin results in a leakage of cytoplasm [17]. A similar mode of action is executed by human beta-defensins (hBDs) [26] and artificially produced peptides, such as artilysine (an amphipathic structure that destabilizes bacteria’s cell walls by hydrolysis) [27]. Furthermore, it has been demonstrated that curcumin is responsible for the down-regulation of 31 quorum-sensing genes required for biofilm production [28]. Additionally, a number of studies have postulated a correlation between curcumin and reduced adherence of *Streptococus mutans* to the tooth surfaces, thereby suppressing biofilm formation [29,30]. In the light of those respective results, these later trials have supported curcumin’s ability to influence multiple signaling pathways and have paved the way for ongoing clinical trials. The respective literature search has proven that there are a remarkable number of clinical investigations focusing on the local application of curcumin administered either as an aqueous solution, gel, chip, or strip [31,32,33,34,35,36,37,38,39,40,41,42,43,44,45,46,47,48,49,50,51,52,53].

Although some meta-analyses have already addressed the issue of curcumin application during periodontal treatment, the present work is not a repetition of previously published results, since most published papers have not dealt with the evaluation of periodontal parameters such as CAL [54,55,56]. Therefore, the overall aim of this systematic review and meta-analysis was to compare the efficacy of scaling and root planing (SRP) alone or in combination with chlorhexidine (CHX) to SRP in combination with local curcumin with respect to key periodontal outcome indicators such as clinical attachment levels and probing pocket depths.

## 2. Materials and Methods

The authors focused on a researchable and answerable study question to the established PICO(ST) format [57]: For adult patients suffering from chronic generalized periodontitis (probing pocket depth of ≥4 mm) (P), will scaling and root planing (SRP), local *Curcuma longa* L. application (I) as compared to SRP or SRP, and local chlorhexidine (CHX) application (C) result in a change of clinical attachment levels (CAL: primary outcome) and probing pocket depths, (PPD: secondary outcome)(O) in a randomized split-mouth design and/or parallel group design studies (S) in a defined period of time (T)? The protocol of the present review was registered at the Prospero Register of Systematic Reviews (registration number: CRD42022290324,10/01/2022; registration name: Effect of locally delivered adjunctive Curcumin in the Treatment of Periodontitis—a Systematic Review and Meta-analysis). The current review was conducted in accordance with the “Preferred Reporting Items for Systematic Reviews and Meta-analyses” (PRISMA) statement checklist [58].

The following databases were searched until August 2022: PubMed, Cochrane Library (Wiley), BASE (base-search.net), LIVIVO, Dentistry Oral Sciences Source (Ebsco), MEDLINE Complete (Ebsco), Scopus, ClinicalTrials.gov, and eLibrary (https://www.elibrary.ru/defaultx.asp; accessed on 3 August 2022.). This combination of information sources retrieved both published journal articles and gray literature (e.g., dissertations, or study register entries). The search strategies were designed by an experienced information specialist (IK). In addition to the search in electronic databases, reference lists of included studies were checked manually. Search results were imported and deduplicated in Endnote 20 (Version 2013; The Endnote Team; Clarivate Analytics, Philadelphia, PA, USA). Additional data for individual search strategies are presented in the Appendix A. The study selection process was performed stepwise. First, two reviewers (LWT and MW) independently screened the titles and abstracts of references found with the literature search. Second, the full texts of the studies included during the previous step were assessed for eligibility. Randomized controlled trials (RCTs) were included that compared SRP alone or in combination with chlorhexidine to SRP and local curcumin/turmeric regarding clinical attachment level and probing pocket depth. Table 1 presents details of the study eligibility criteria.

The quality of the included trials was methodically assessed by two authors (LWT and MW) using the revised Cochrane risk of bias tool 2.0 for randomized trials (RoB2). Any possible dissensions were resolved by discussion and mutual agreement. RoB2 is arranged into five different disciplines (randomization process, deviations from intended interventions, missing outcome data, measurement of the outcome, and selection of the reported results) that aim to evaluate all aspects of the study that are related to the risk of bias [59]. The five different disciplines were judged as having either low risk, some concerns, or high risk, and according to this judgment, an overall assessment of the risk of bias level in each individual study was made.

One author (LWT) collected the relevant data from the included articles. This was cross-checked for accuracy and completeness by another author (MW). The data of interest were methodology, number of participants, participants baseline characteristics, concentration of curcumin/turmeric and chlorhexidine, evaluation period, and results for primary and secondary outcomes. The authors of the Dave et al. (2018) study were contacted to gather missing information concerning initial PPD and curcumin concentration.

A random-effects meta-analysis was performed using an inverse-variance model with the DerSimonian–Laird estimate of squared tau (τ^2^) by pooling mean differences with 95% confidence intervals, if the number of identified investigations that were similar in population and outcome was sufficient. The statistical heterogeneity was assessed across trials by visually inspecting the forest plots and calculating the I^2^ statistics. STATA release 17.0 was used for all analyses (StataCorp LLC; College Station, TX, USA). Additional calculations had to be performed for the data published by Raghava et al. (2019) and Farhood et al. (2020); CAL was presented in variance (PPD in standard error), and CAL and PPD were presented in standard error, respectively. Conversions were made to the standard deviation.

## 3. Results

### 3.1. Literature Search and Screening

A total of 827 records were identified through the literature search. After deduplication, 222 studies were screened by title and abstract. Consecutively, 33 full-text articles were assessed for eligibility, and, finally, 23 investigations were included [31,32,33,34,35,36,37,38,39,40,41,42,43,44,45,46,47,48,49,50,51,52,53]. Details of the study selection process are presented in Figure 1.

### 3.2. Study and Patient Characteristics

Sixteen studies compared SRP with SRP and local curcumin/turmeric application [31,32,33,34,35,36,37,38,39,40,41,42,43,44,45,46], three investigations compared SRP and chlorhexidine application with SRP and local curcumin/turmeric application [47,48,49], while another four studies evaluated SRP and SRP in combination with CHX compared to SRP and local curcumin/turmeric [50,51,52,53]. While a split-mouth design was applied in 18 investigations [31,32,33,34,35,38,40,41,43,44,45,46,47,48,49,50,51,53], five investigations employed a parallel group design [36,37,39,42,52]. The number of participants in the included studies ranged from 10 to 90. The age of participants ranged from 20 to 65 years; the mean age or gender ratio could not be calculated due to missing uniform data concerning these variables among certain studies.

Included were studies carried out at university hospitals in India, Iraq, Egypt, and Brazil, where participants were recruited from the departments of periodontology. Study duration ranged from 21 days to 3 months. Thirteen studies used an acrylic stent to measure probing pocket depth and/or clinical attachment level [31,32,33,34,39,41,43,45,46,47,51,52,53]. Fourteen investigations used a COE pack to ensure the duration of the applied medicament [31,32,33,35,38,39,41,42,43,45,48,49,50,52]. The periodontal condition requiring treatment was defined as a probing pocket depth of ≥5 mm in ten studies [34,38,39,40,43,45,46,47,49,53], between 5 and 7 mm in seven [31,33,35,36,41,50,52], and between 4 and 6 mm in two investigations [37,48]. The following initial PPDs were observed once: between 5 and 8 mm [51], between 5 and 6 mm [44], >5 mm [32], and ≥4 mm [42]. Three studies were included that solely reported PPD [36,44,53]. Details of study characteristics are summarized and presented according to their controls and interventions in Table 1.

**Table 1 biomedicines-11-00481-t001:** Characteristics of included studies. SMD: split-mouth design, PGD: parallel-group design PPD: probing pocket depth, SBI: sulcus bleeding index, PI: plaque index, CAL: clinical attachment level, SRP: scaling and root planning, GI: gingival index, BOP: bleeding on probing, CHX: chlorhexidine [31,32,33,34,35,36,37,38,39,40,41,42,43,44,45,46,47,48,49,50,51,52,53].

Study	Sample Size	Study Design	Clinical Parameters	Intervention and Control				Stent	COE Pack	Follow-Up Periods
**Studies Comparing SRP alone to SRP and Curcumin**										
Behal et al., 2011 [31]	*n* = 30 PPD 5–7 mm	SMD	PI (Turesky-Gilmore-Glickman), GI (Löe and Silness), SBI (Muhlemann), PPD (William probe), CAL	Group 1: SRP alone		Group 2: SRP was followed by local application 2% turmeric gel		yes	yes	0, 30, 45 d /0, 30 d
Bhatia et al., 2014 [32]	*n* = 25, (15♂, 10♀, 21–45 y.o.) PPD > 5 mm.	SMD	PI (Silness and Löe), SBI (Muhlemann), PPD, CAL	Group 1: SRP alone		Group 2: SRP was followed by local application of 1% curcumin gel		yes	yes	0, 1, 3, 6 m /0, 1 m
Anuradha et al., 2015 [33]	*n* = 30 (25–60 y.o.) PPD 5–7 mm	SMD	PI (Turesky-Gilmore-Glickman), GI (Löe and Silness), PPD, CAL (UNC-15)	Group 1: SRP alone		Group 2: SRP was followed by local application of curcumin gel (10 mg of *Curcuma longa* extract/g)		yes	yes	0, 30, 45 d /0, 30 d
Nagasri et al., 2015 [34]	*n* = 30 (12♂, 18♀, 35–60 y.o.) PPD ≥ 5 mm	SMD	PI (Silness and Löe), GI (Löe and Silness), PPD, CAL	Group 1: SRP alone		Group 2: SRP was followed by local application of curcumin gel (10 mg of *Curcuma longa* extract/g)		yes	no	0, 4 w /0, 4 w
Shivanand et al., 2016 [35]	*n* = 14 (6♂, 8♀, 35–50 y.o.) PPD 5–7 mm	SMD	PI (Silness and Löe), GI (Löe and Silness), BOP, PPD, CAL	Group 1: upper arch received SRP alone		Group 2: lower arch received SRP and application of curcumin gel (10 mg of *Curcuma longa* extract/g)		no	yes	0, 21, 30, 90 d /0, 30 d
Nasra et al., 2017 [36]	*n* = 10 (35–55 y.o.) PPD 5–7 mm	PGD	PPD (Glavind and Löe), SBI (Checchi), GI (Löe and Silness), PI (Silness and Löe)	Group 1: SRP alone		Group 2: SRP and curcumin gel 2%, the application was repeated once weekly over three weeks period		no	no	0, 1 m /0, 1 m
Dave et al., 2018 [37]	*n* = 20 (9♂, 11♀, 20–59 y.o.) PPD 4–6 mm	PGD	PI, BOP, SBI, PPD, CAL (UNC15)	Group1: SRP alone		Group 2: SRP and curcumin gel 10%, patients were instructed to apply gel for 2–3 min once daily		no	no	0, 2 m /0, 2 m
Raghava et al., 2019 [38]	*n* = 10 (5♂, 5♀, 25–40 y.o.) PPD ≥ 5 mm	SMD	PI (Silness and Löe), GI (Löe and Silness), PPD, CAL	Group 1: SRP alone		Group 2: SRP was followed by local application of curcumin gel (10 mg of *Curcuma longa* extract/g)		no	yes	0, 4 w /0, 4 w
Kaur et al., 2019 [39]	*n* = 29 (20♂, 9♀, 20–65 y.o.) PPD ≥ 5 mm	PGD	PI (Silness and Löe), SBI (Muhlemann), PPD, CAL (UNC-15)	Group 1: SRP alone		Group 2: SRP was followed by local application 1% curcumin gel (10 mg of *Curcuma longa* extract/g)		yes	yes	0, 1,3 m /0, 1 m
Perez-Pacheco et al., 2020 [40]	*n* = 20 (6♂, 14♀, 37–62 y.o.) PPD ≥ 5 mm	SMD	PI (O’Leary), GI (Ainamo and Bay), BOP, PPD, gingival recession, CAL (UNC-15)	Group 1: SRP and 0.05 mg/mL nano capsulated curcumin		Group 2: SRP and empty nanoparticles		no	no	0, 1, 2, 6 m /0, 1 m
Farhood et al., 2020 [41]	*n* = 20 (9♂, 11♀, ≥21–45 y.o.) PPD 5–7 mm	SMD	PI, GI, BOP, PPD, CAL	Group 1: SRP alone		Group 2: SRP was followed by local application of curcumin gel (10 mg of *Curcuma longa* extract/g) Second application after 1 week		yes	yes	0, 1 m /0, 1 m
**Studies comparing SRP alone to SRP and Curcumin and a third or fourth control (not included in this meta-analysis)**										
Mohammed et al., 2020 [42]	*n* = 90 (35♂, 55♀, 25–54 y.o.) PPD ≥ 4 mm	PGD	PPD, CAL, GI, BOP	Group 1: healthy periodontium (control group)		Group 2: for periodontitis patients SRP and curcumin gel (C. Longa extract, 10 mg)	Group 3: periodontitis patients receiving SRP alone	no	yes	0, 1 m /0, 1 m
Rahalkar et al., 2021 [43]	*n* = 15 (5♂, 10♀, 37–57 y.o.) PPD ≥ 5 mm	SMD	PPD, CAL, GI (Löe and Silness), PI (Silness and Löe), SBI (Mombelli)	Group 1: SRP alone		Group 2: SRP and curcumin gel (C. Longa extract, 10 mg)	Group 3: SRP and tulsi extract	yes	yes	0, 30 d /0, 30 d
Elavarasu et al., 2016 [44]	*n* = 15 (35–50 y.o.) PPD 5–6 mm	SMD	PI, GI, SBI, PPD	Group 1: Healthy periodontium (control)		Group 2: SRP alone	Group 3: SRP and curcumin strip placement 0,2% loaded on to guided tissue membrane (GTR)	no	no	0, 21 d /0, 21 d
Saini et al., 2021 [45]	*n* = 30 (30–65 y.o.) PPD ≥ 5 mm	SMD	PI, GI, PPD, CAL (UNC-15)	Group 1: SRP and 5% neem chip		Group 2: SRP and 5% turmeric chip	Group 3: SRP and placebo chip	yes	yes	0, 1, 3 m /0, 1 m
Sreedhar et al., 2015 [46]	*n* = 15 (15P, 7♂, 8♀, 35–55 y.o.) PPD ≥ 5 mm	SMD	PI, SBI, PPD, CAL	Group 1: SRP alone.	Group 2: SRP anfd curcumin gel (10 mg of *Curcuma longa* extract/g) application for 5 min	Group 3: SRP and curcumin application for 5 min and irradiation with blue light emitting diode	Group 4: SRP and curcumin PDT	yes	no	0, 1, 3 m /0, 3 m
**Studies comparing SRP and CHX to SRP and Curcumin**										
Gottumukkala et al., 2014 [47]	*n* = 60 (25–55 y.o.) PPD ≥ 5 mm	SMD	PPD, CAL, GI (Löe and Silness), PI (Silness and Löe)	Group 1: SRP alone along with CHX chip (2.5 mg)		Group 2: received SRP along with curcumin chip (CU extract concentration of 50 mg/cm)		yes	no	0, 1, 3, 6 m /0, 1 m
Anitha et al., 2015 [48]	*n*= 30 (20♂, 10♀, 20–50 y.o.) PPD 4–6 mm	SMD	PPD, CAL, GI (Löe and Silness), PI (Turesky-Gillmore)	Group 1: receiving SRP and curcumin gel (250 g of the powdered rhizome of *Curcuma longa* in 5 mL ethanol) Repeated application at day 15		Group 2: SRP and CHX gel 0.1% Repeated application at day 15		no	yes	0, 15, 30 d /0, 30 d
Siddarth et al., 2020 [49]	*n*= 25 (20♂, 5♀, ≥30 y.o.) PPD ≥ 5 mm	SMD	GI (Löe and Silness), PI (Silness and Löe), SBI (Muhlemann), PPD, CAL	Group 1: SRP and application of 2% curcumin gel		Group 2: SRP and application of 0.2% CHX gel		no	yes	0, 1, 3 m /0, 1 m
**Studies comparing SRP alone to SRP and Curcumin and SRP and CHX**										
Jaswal et al., 2014 [50]	*n* = 15 (12♂, 3♀, 21–55 y.o.) PPD 5–7 mm	SMD	PI (Silness and Löe), GI (Löe and Silness), PPD, CAL (UNC-15)	Group 1: received SRP and 2% turmeric gel		Group 2: receiving SRP and 1% CHX	Group 3: SRP alone	no	yes	0, 30, 45 d /0, 30 d
Singh et al., 2018 [51]	*n* = 40 (22♂, 18♀, 30–50 y.o.) PPD 5–8 mm	SMD	PI, GI, PPD, CAL (UNC-15)	Group 1: SRP and sites treated with CHX chip (2.5 mg)		Group 2: SRP and sites treated with 5% turmeric chip	Group 3: SRP alone	yes	no	0, 1, 3 m /0, 1 m
Guru et al., 2020 [52]	*n* = 45 (36♂, 9♀, 25–50 y.o.) PPD 5–7 mm	PGD	PI, GI, PPD, CAL (UNC-15)	Group 1: SRP alone		Group 2: SRP, 2%curcumin with nanogel	Group 3: SRP, 1% CHX gel	yes	yes	0, 21, 45 d /0, 21 d
Gottumukkala et al., 2013 [53]	*n* = 26 (12♂, 14♀, 30–55 y.o.) PPD ≥ 5 mm	SMD	PI (Silness and Löe), BOP, redness, PPD (UNC15)	Group 1: SRP and saline irrigation Repeated gingival irrigation at 7,14 and 21 d		Group 2: SRP and 1% curcumin solution, repeated gingival irrigation at 7, 14 and 21 d	Group 3: SRP and 0.2% CHX Repeated gingival irrigation at 7, 14 and 21 d	yes	no	0, 1, 3, 6 m /0, 1 m

### 3.3. Quality Assessment

The possibility of bias in design and analysis was evaluated by the Cochrane Risk of Bias tool 2.0 [59] (this tool was designed for randomized parallel group design investigations, and, consequently, this should be carefully considered when interpreting the following results). In total, 23 studies were rated with a moderate risk of bias. The most common source of potential bias was domain four (“measurement of the outcome”). In all included studies, investigators probed manually, which was judged to have a poor validity. The second most common source of bias was domain three (“missing outcome data”); many investigations [31,32,33,34,35,38,41,42,43,44,45,46,47,48,49,50,51,53] failed to comment on the loss of follow-up, which led to the present judgment. The third most common source of bias was the “randomization process”, which certain investigations could have described it in greater detail [31,32,35,36,41,42,43,44,50,53]. The remaining domains, “effect of assigning to intervention”, as well as “selection to reported results”, performed acceptably throughout the included investigations. Details of the RoB Assessment are provided in Figure 2.

### 3.4. Clinical Attachment Level Loss

#### 3.4.1. SRP Alone Compared to SRP and Local Curcumin

Seventeen investigations [31,32,33,34,35,37,38,39,40,41,42,43,45,46,50,51,52] evaluated the effects of SRP and curcumin/turmeric on the loss of CAL. In random effects meta-analysis, a statistically significant mean difference of −0.33 mm (95% CI −0.54 to −0.11; *p* = 0.03, I^2^ = 62.3%; 453 sites; see Figure 3) was favoring SRP and curcumin/turmeric was observed.

#### 3.4.2. SRP and Chlorhexidine Compared to SRP and Local Curcumin

The effect of SRP and chlorhexidine application, in comparison to SRP and local curcumin/turmeric application, on the loss of CAL was evaluated by six investigations [47,48,49,50,51,52]. Random effects meta-analysis showed a statistically non-significant mean difference of −0.42 mm (95% CI −1.15 to 0.31; *p* = 0.258, I^2^ = 93.6%; 185 sites; see Figure 4) in favor of SRP and curcumin/turmeric.

### 3.5. Probing Pocket Depth Reduction

#### 3.5.1. SRP Alone Compared to SRP and Local Curcumin

Twenty studies with a moderate risk of bias [31,32,33,34,35,36,37,38,39,40,41,42,43,44,45,46,50,51,52,53] reported on the difference in probing pocket depth reduction between SRP alone and SRP with local curcumin/turmeric application. Random effects meta-analysis showed a statistically significant difference of −0.47 mm (95% CI −0.88 to −0.06; *p* = 0.024; I^2^ = 95.5%; 501 sites; see Figure 5).

#### 3.5.2. SRP and Chlorhexidine Compared to SRP and Local Curcumin

Figure 6 presents the effects of seven investigations [47,48,49,50,51,52,53] comparing SRP and chlorhexidine application, in contrast to SRP and curcumin/turmeric application on probing pocket depth reduction. Random effects meta-analysis exhibited a pooled mean difference of −0.54 mm (95% CI −1.19 to 0.12; *p* = 0.108, I^2^ = 93.4%, 208 sites; see Figure 6), which was statistically non-significant.

## 4. Discussion

This systematic review and meta-analysis revealed a statistically significant difference between SRP alone, compared to SRP and curcumin/turmeric application for CAL and PPD. However, these significant decreases (CAL −0.33 mm; PPD −0.47 mm) are not considered clinically relevant. Included studies showed a notable degree of heterogeneity, which is probably due to different application methods, varying concentrations of curcumin/CHX, and the unreliability of outcome measurements.

All periodontal pockets were probed manually by all investigators, even though one investigation used a pressure-sensitive manual probe [53] and two investigations measured six sites around each tooth [40,43]. It is well accepted that probing manually might be unreliable, and can result in certain inaccuracies [61,62], at least to some extent. A standardized measuring method using electronic periodontal probes, thus controlling probing force and measuring to the closest tenth of a millimeter and, therefore, generating reproducible results, even with different examiners, would seem generally preferable [63,64]. Furthermore, factors like the design of the probe, probing position, visual observational error, and tissue inflammation could influence the reproducibility of readings, thus leading to detection bias [65,66]. Nevertheless, the assessors of the present investigation decided to override the suggested overall judgment for the risk of bias assessment. This is justified by the fact that all treatment groups measured the outcomes manually, in most cases uniformly using a UNC 15 periodontal probe [33,37,39,40,45,50,51,52,53]. Additionally, studies have shown that there is a tendency to have similar reliability between manual and electronic probes [67,68]. Furthermore, many investigations [31,32,33,34,35,38,41,42,43,44,45,46,47,48,49,50,51,53] failed to comment on the loss of follow-up, and this sort of attrition bias led to the present judgment.

A further plausible explanation for the observed heterogeneity might be the varying application methods. To a certain extent, gels have several advantages over other application methods. They are biocompatible and bioadhesive, allowing them to attach to periodontal pockets and, furthermore, enabling a controlled drug release and minimum dose frequency [69]. While most studies applied a conventional gel [32,33,34,35,36,37,39,40,41,42,43,44,45,48,50,51], two used a nanoparticle gel system, which is expected to have several advantages. Due to their nanoparticle size, these gel systems are able to penetrate into the most apical regions of periodontal pockets, thus ensuring a homogenous disposal of the drug over a long interval, along with a reduction in drug quantity and high bioavailability [69]. Matrix delivery systems such as chips and strips, as used in four trials [46,47,49,52], have the benefit of sustained drug release patterns. In contrast, solutions, as used in one investigation [54], provide high concentrations initially and will be diluted promptly by liquids, for example, by gingival crevicular fluid [69].

There are numerous issues to be discussed about the stated concentrations in the included investigations. Certain studies used different degrees of purity as starting products. The majority of selected studies, nine in total, used the product “Curenext oral gel” (Abbott Healthcare Limited, Mumbai, India), which contains 10 mg of *Curcuma longa* extract/g [34,35,36,37,39,42,43,44,45]. The company (Abbott Healthcare) was contacted, and they stated that “*Curcuma longa* extract” contains 72.77% curcumin. Turmeric contains, apart from curcumin and curcumin’s analogs, several other substances and phytochemicals, such as zingiberene, eugenol, turmerin, turmerones, and turmeronols [70]. Additionally, different manufacturing processes were used to yield curcumin products. One study used a 95% pure curcumin powder to create a 2% curcumin gel [36], while another investigation purified curcumin by the method of evaporation, thus generating 99% curcumin powder [49]. A further investigation used a >65% curcumin powder to create curcumin nanoparticles [40,71], whereas another investigation estimated the curcumin content by measuring the absorbance of curcumin spectrophotometrically [32]. The starting product of one study (using >10% curcumin) might have contained a range of other substances [52,72]. The remaining investigations did not describe the initial product used in sufficient detail, and the amount of curcumin was not specified [31,37,44,45,48,50,51,53]. Additionally, varying CHX concentrations were noted: 0.1% [48], 0.2% [49,53], 1% [50,52], and 2.5 mg [47,51].

Although the studies under investigation all revealed varying curcumin concentrations, it can be assumed that the latter was effective, since an inhibitory effect on 61.01% of the MMP-9 activity has been determined at a curcumin concentration of 1500 μg/mL [73]. Additionally, a few investigations repeated the curcumin application. Patients were either instructed to apply the gel daily [37], or the application was repeated once weekly (over a three-week period) [36], with a second application of gel after one week [41], repeated application at day 15 [48], or repeated applications after 7, 14, and 21 days [53]. Furthermore, the use of a COE pack after drug application, aiming to ensure the persistence of the applied drug and prevent site contamination, could be a positive influencing factor for the concentration of the drug applied; however, this has not been proven clinically up to now.

Previous trials have commented on curcumin’s poor bioavailability, and this was probably due to low absorption, rapid metabolism, and quick systemic elimination [74,75]. To date, a number of investigations have begun to examine the use of synthetic, structural analogues and various nanoforms of curcumin, as they have improved plasma and tissue levels [74,75,76]. The effect of oral application of curcumin and modified curcumin on bone resorption, inflammation, and apoptosis in rats, was compared by a previous study. It was concluded that administration of chemically modified curcumin significantly reduced the inflammatory infiltrate in comparison to natural curcumin [77]. This should encourage interest in planning and conducting trials to explore the qualities of chemically modified curcumin.

Limitations of this work are that the risk of bias assessment used was based on a study design used in general medicine. There seems to be a lack of risk of bias in assessments designed for split-mouth trials, which is a common design in oral health. It is worth noting that, even though a split-mouth design can be considered powerful, this latter methodology may result in considerable variability, which is a result of characteristic differences between examiners. No doubt, and this cannot be ruled out, there was a potential risk of contamination of the control site, as it could be possible for the drugs to diffuse to the other site (carry-across effects) [78]. In comparison to the protocol registration, a few amendments would seem worth mentioning. No subgroup analysis was conducted; more data bases were searched; the search was updated in April 2022, and a slight alteration was made to the title; the STATA release 17.0 was used for all analyses (StataCorp LLC; College Station, TX, USA).

The search for new anti-microbial and anti-inflammatory substances as potential agents in the treatment of oral diseases, especially those that cannot develop antibiotic resistance, has gained much importance in recent years. Curcumin appears to possess these valuable properties and has reached the clinical testing phase. At first glance, the results of these clinical studies appear very promising. On a closer inspection, however, it must be stated that several issues in relation to the risk of bias, as discussed in detail, must be elucidated. Uniform study designs and methods with accurate and reproducible measurements of endpoints and homogenous concentrations would be desirable. Interestingly, a recently published meta-analysis of the effect of adjuvant curcumin in the treatment of periodontitis, came to the conclusion that curcumin can be successfully used in periodontal therapy [57]. Another meta-analysis concerning this topic, evaluating gingival index, sulcus bleeding index, and bleeding on probing as primary outcomes, concluded that curcumin is a “good candidate as an adjunct treatment for periodontal disease” [56]. Another meta-analysis concludes that locally applied curcumins “were found to be equally effective compared to the routinely used agents for reduction of plaque and gingival inflammation” [55]. Undoubtedly, such statements should be carefully but critically weighed; when reflecting on several aspects, like the lack of sufficient details concerning study quality, this would seem more than justified. Moreover, to recommend curcumin for periodontal therapy would call for clear endpoints assessing the effectiveness of curcumin in periodontal treatment, and any possible effects of the oral hygiene of participants must be clearly distinguished.

When pondering on the treatment of periodontal disease with adjuvant curcumin, the data available from this present meta-analysis would suggest that there is obviously no reason for any further investigations, and this would refer both to large-scale and high-quality studies. At the end of the day, the available data base could reveal that the proven (and noteworthy, no doubt) biochemical properties of curcumin would justify the previous research projects in the first instance, but, notwithstanding, no clinically significant improvements could be proven with the current systematic review. Consequently, a clinical implementation of adjuvant curcumin for periodontal treatment is not recommended.

## 5. Conclusions

In conclusion, with reference to clinical attachment level and probing pocket depth, the present results cannot indicate that curcumin’s/turmeric’s anti-bacterial and anti-inflammatory properties result in an additionally beneficial clinical outcome, when combining this adjunct to scaling and root planing. Therefore, our findings do not support the application of curcumin-/turmeric-based products in non-surgical periodontal treatment scenarios.

## Figures and Tables

**Figure 1 biomedicines-11-00481-f001:**
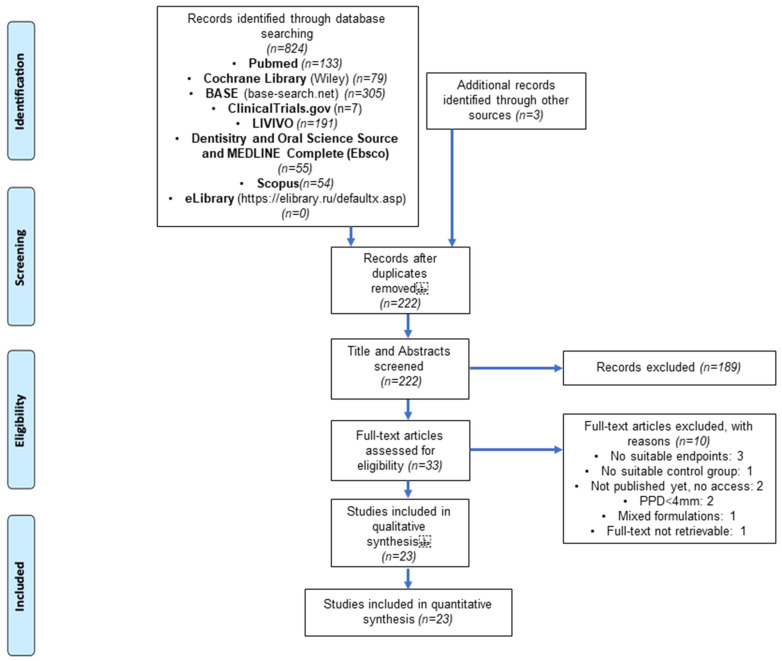
PRISMA Flow Diagram modified from Moher et al. (2009) [60], showing the number of studies identified, screened, assessed for eligibility, excluded, and included in the systematic research [31,32,33,34,35,36,37,38,39,40,41,42,43,44,45,46,47,48,49,50,51,52,53].

**Figure 2 biomedicines-11-00481-f002:**
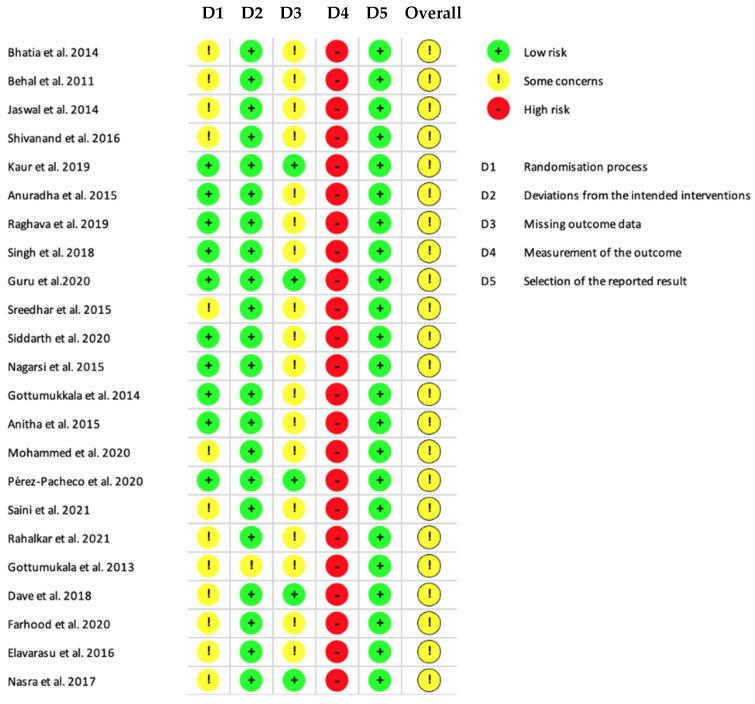
Diagram showing the bias detected in the different domains and overall bias of studies included [31,32,33,34,35,36,37,38,39,40,41,42,43,44,45,46,47,48,49,50,51,52,53]. Based on Sterne et al. (2019) [59].

**Figure 3 biomedicines-11-00481-f003:**
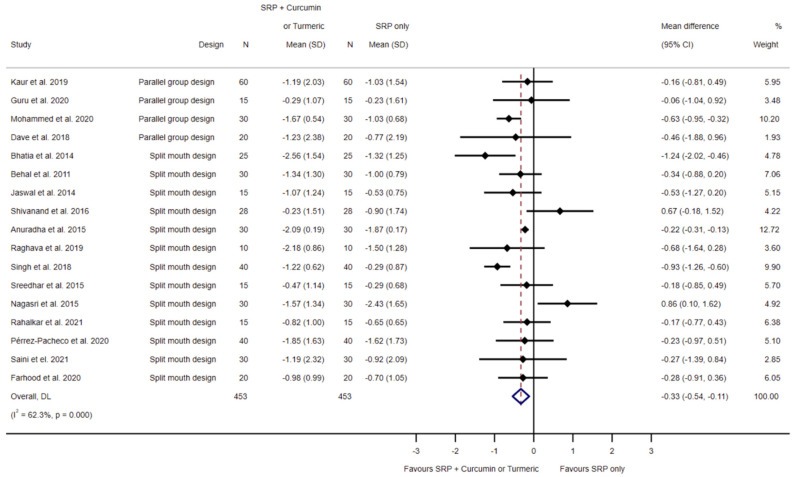
Forest plot for the comparison of SRP to SRP and local curcumin/turmeric application related to clinical attachment level loss [31,32,33,34,35,37,38,39,40,41,42,43,45,46,50,51,52].

**Figure 4 biomedicines-11-00481-f004:**
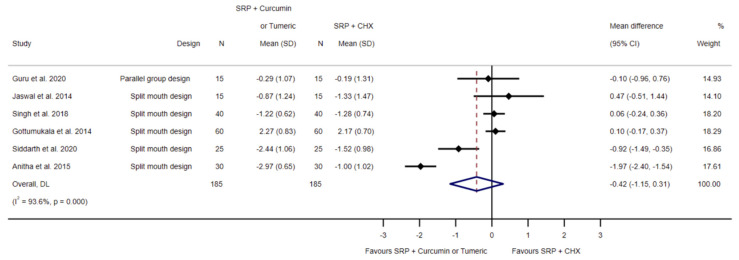
Forest plot for the comparison of SRP and chlorhexidine to SRP and local curcumin/turmeric related to clinical attachment level loss [47,48,49,50,51,52].

**Figure 5 biomedicines-11-00481-f005:**
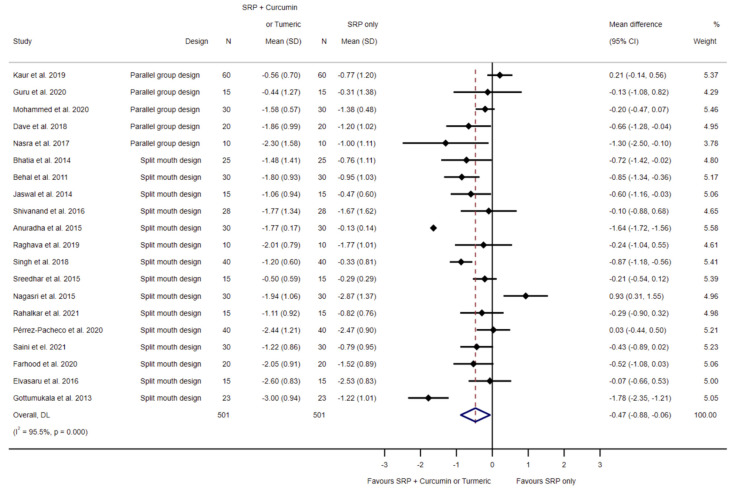
Forest plot for the comparison of SRP to SRP and local curcumin/turmeric application related to probing pocket depth reduction [31,32,33,34,35,36,37,38,39,40,41,42,43,44,45,46,50,51,52,53].

**Figure 6 biomedicines-11-00481-f006:**
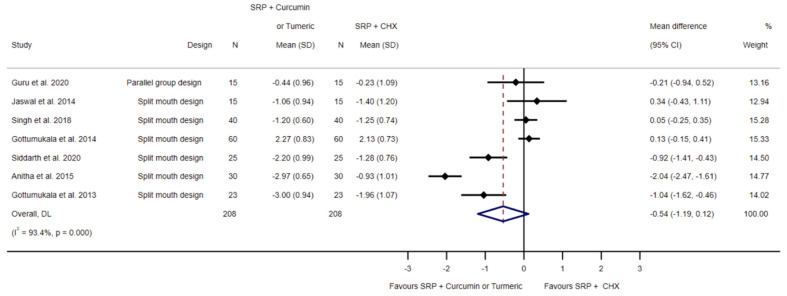
Forest plot for the comparison of SRP and chlorhexidine to SRP and local curcumin/turmeric application related to probing pocket depth reduction [47,48,49,50,51,52,53].

## Data Availability

Datasets generated in this study can be found in the Appendix A.

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
