# Peer review of "A Systematic Review and Meta-Analysis on the Efficacy of Locally Delivered Adjunctive Curcumin (Curcuma longa L.) in the Treatment of Periodontitis"

_biomedicines, 2023, doi:10.3390/biomedicines11020481_

Round 1

Reviewer 1 Report

The study is well designed.

Author Response

We are very grateful for the reviewers’ recommendations provided by the editors. The comments are encouraging, and the reviewers appear to share our judgement that this meta-analysis and its results are clinically important. Please see below, in italics, our detailed responses to the comments. All page and line numbers refer to the manuscript file with tracked changes (highlighted yellow).

First Reviewer

  1. The study is well designed.

We greatly appreciate this reassuring comment.

Once again, we would wish to sincerely thank ALL reviewers. No doubt, with your comments, you have spent enormous time to voluntarily review our draft, and this has resulted in a considerable improvement of this paper.

Reviewer 2 Report

In Introduction main features of curcumin and emerging research direction  should be added and related reference added such as: ZieliÅ„ska et al. 2020. doi:10.3390/medicina56070336.

The novelty character of paper should be marked.

Table 2 should be better explain in the text.

The paragraph Quality assessment should be better described and implemented.

A Section Conclusion should be inserted

Author Response

We are very grateful for the reviewers’ recommendations provided by the editors. The comments are encouraging, and the reviewers appear to share our judgement that this meta-analysis and its results are clinically important. Please see below, in italics, our detailed responses to the comments. All page and line numbers refer to the manuscript file with tracked changes (highlighted yellow).

Second Reviewer

  1. In Introduction main features of curcumin and emerging research direction should be added and related reference added such as: Zielińska et al. 2020. Doi:10.3390/medicina56070336.

Thank you very much for this remark; the statement was modified in accordance with your suggestion (page 2, lines 64-65; page 16, lines 425-427).

  1. The novelty character of paper should be marked.

We are grateful for this comment; with the “Introduction” section (page 2, lines 87-94), we have tried to emphasize the novelty character of this meta-analysis, and the relevant information has been added on page 14, lines 351-355.

  1. Table 2 should be better explained in the text.

We thank the reviewer for this comment. To facilitate the interpretation of Table 2 (now Table 1), we have divided the latter into subsections with separate titles. In addition, we also added the following sentences to the results section, to additionally clarify interpretation of the Table (page 5, lines 181-183).

  1. The paragraph Quality assessment should be better described and implemented.

We appreciate your comment, no doubt. The relevant information has been added now, please see page 3, lines 132-134. Furthermore, additional aspect has been described with Figure 2, to facilitate interpretation (please see page 9, line 204).

  1. A Section Conclusion should be inserted.

A conclusion has been inserted on page 14, lines 365-371.

Once again, we would wish to sincerely thank ALL reviewers. No doubt, with your comments, you have spent enormous time to voluntarily review our draft, and this has resulted in a considerable improvement of this paper.

Reviewer 3 Report

Dear authors,

It was a pleasure to read your study entitled “A systematic review and meta-analysis on efficacy of locally delivered adjunctive Curcumin (Curcuma Longa L.) in the treatment of periodontitis.” As a reviewer, I have some suggestions and questions about your manuscript.

You must send the manuscript to a proofreading service. 

- Introduction

Remove the hypothesis from the last paragraph. 

- Methods and Results

The study’s question is not straightforward in this section. Therefore, please, include the question showing the PICOS strategy, e.g., What is XXXXXXX(P)XXXXXXXX(I)XXXXXX (C) XXXXXXXX(O) XXXXXXXX (S) XXXXX(T)?)

“Curcumin in the Treatment of Periodontitis − a Systematic Review and Metaanalysis) of the present review was pre-registered at the Prospero Register of Systematic

Reviews (PROSPERO)” - Pre-registered? Please change it to registered and move the number to the same sentence.

Table 1 confirmed that you used a different type of search strategy. However, you described in the methods that you followed the PICO(S). Therefore, in Table 1, it is possible to observe the use of your Research Question based on PICOST (Population, Intervention, Control, Outcomes, Study design, and Timeframe). Please, include the correct information in your study.

Please, include the PRISMA Flow Diagram in your methods.

Also, finding the same number of papers for qualitative and quantitative analysis is interesting. However, you saw in the Cochrane Risk of Bias tool 2.0 all the studies with a high risk in the measurement of the outcome.

What were the criteria for creating Table 2? First, the authors need to put the studies in a chronological sequence.

If you use the Cochrane risk of bias tool 2.0 for your analysis, why did you not use the RevMan5 for the Metanalysis?

Review the results in “SRP alone compared to SRP and local curcumin” according to the Forest plot; it does not look like you found a significant difference. 

- Discussion 

The study is not a new theme. So why did you choose only two previous SRs included in your discussion? And what were the criteria for doing that?  

Author Response

We are very grateful for the reviewers’ recommendations provided by the editors. The comments are encouraging, and the reviewers appear to share our judgement that this meta-analysis and its results are clinically important. Please see below, in italics, our detailed responses to the comments. All page and line numbers refer to the manuscript file with tracked changes (highlighted yellow).

Third Reviewer

It was a pleasure to read your study entitled “A systematic review and meta-analysis on efficacy of locally delivered adjunctive Curcumin (Curcuma Longa L.) in the treatment of periodontitis.” As a reviewer, I have some suggestions and questions about your manuscript.

You must send the manuscript to a proofreading service.

Thank you for your recommendation. We have followed your suggestion, and a thorough proofreading has considerably improved readability of our draft.

  1. Remove the hypothesis from the last paragraph.

The hypothesis was removed.

  1. Methods and Results. The study’s question is not straightforward in this section. Therefore, please, include the question showing the PICOS strategy, e.g., What is XXXXXXX(P)XXXXXXXX(I)XXXXXX © XXXXXXXX(O) XXXXXXXX (S) XXXXX(T)?)

This. indeed, would seem a relevant addition. PICOS strategy was described according to your example (please see page 3, lines 97-103); consequently, Table 1 has been removed.

  1. “Curcumin in the Treatment of Periodontitis − a Systematic Review and Metaanalysis) of the present review was pre-registered at the Prospero Register of SystematicReviews (PROSPERO)” - Pre-registered? Please change it to registered and move the number to the same sentence.

Wording has been changed to “registered”, and the registration number has been moved (page 3, lines 103-104).

  1. Table 1 confirmed that you used a different type of search strategy. However, you described in the methods that you followed the PICO(S). Therefore, in Table 1, it is possible to observe the use of your Research Question based on PICOST (Population, Intervention, Control, Outcomes, Study design, and Timeframe). Please, include the correct information in your study.

The correct information has been included now (please see page 3, lines 97-103). Thank you very much for your vigilance.

  1. Please, include the PRISMA Flow Diagram in your methods.

We regret to disagree with this comment. According to Cochrane Handbook, the PRISMA Flow diagram should be placed in the results section (please see Section III.3.5.1. https://training.cochrane.org/handbook/current/chapter-iii). We hope that you will agree.

  1. Also, finding the same number of papers for qualitative and quantitative analysis is interesting. However, you saw in the Cochrane Risk of Bias tool 2.0 all the studies with a high risk in the measurement of the outcome.

When looking at Figure 2, it is indeed noticeable that the highest risk level was assigned to area D4 "Measurement of the outcome". Please note that this was assigned because the authors of all previous studies used manual measurements of periodontal pockets. We believe that measuring periodontal pockets with uncalibrated methods, such as a simple periodontal probe, clearly represents a considerable weakness of the study design. When looking at the results, it is also noticeable that the change in periodontal pockets is sometimes in the range of less than one millimeter. Such changes cannot be objectively assessed using manual evaluation methods. Electronic calibrated probes, such as the Florida probe, have become widely used for this purpose, but, unfortunately, they have not been used with any of the included studies. Consequently, this would mean that not a single study can be included in our meta-analysis. Notwithstanding, due to this fact, we would like to emphasize that the conclusions of all previous meta-analyses published to date would seem worthless or should clearly be scrutinized (we would suppose you agree). All these meta-analyses, however, do include the same studies, and only the parameters that have been examined and summarized are different.

  1. What were the criteria for creating Table 2? First, the authors need to put the studies in a chronological sequence.

Table 2 was created for the reader to provide an overview for different characteristics of the various studies included in this meta-analysis. The studies have been subdivided according to their intervention and control. To facilitate interpretation, titles have been added to subsections.

  1. If you use the Cochrane risk of bias tool 2.0 for your analysis, why did you not use the RevMan5 for the Metanalysis?

For meta-analysis we preferred to use STATA, since this statistical software provides more flexibility regarding presentation of forest-plots than RevMan 5.

  1. Review the results in “SRP alone compared to SRP and local curcumin” according to the Forest plot; it does not look like you found a significant difference.

For comparison of SRP alone and SRP plus local curcumin application we observed a small but statistically significant difference regarding both, clinical attachment level loss (pooled mean difference -0.33 mm; 95% CI -0.54 to -0.11; p = 0.03; see Figure 3), and probing pocket depth reduction (pooled mean difference -0.47 mm; 95% CI -0.88 to -0.06; see Figure 5). However, as outlined in the manuscript, we do not consider these decreases clinically relevant. For consistency reasons, we have added the p-value to the effect estimates for probing pocket depth reduction.

  1. The study is not a new theme. So why did you choose only two previous SRs included in your discussion? And what were the criteria for doing that?

Thank you for this question. We appreciate the opportunity to include one further meta-analysis in our Discussion section (please see page 14, lines 349-355). Further meta-analyses indeed have focused on other, in our opinion, less meaningful parameters, such as bleeding on probing, plaque accumulation or gingival recession and were therefore not include as a comparison in the discussion. To the best of our knowledge, there is no review of PPD and CAL to date.

Once again, we would wish to sincerely thank ALL reviewers. No doubt, with your comments, you have spent enormous time to voluntarily review our draft, and this has resulted in a considerable improvement of this paper.

Round 2

Reviewer 3 Report

Thank you for your answers and modifications in the manuscript.